# Application of Machine Learning Methods for Epilepsy Risk Ranking in Patients with Hematopoietic Malignancies Using

**DOI:** 10.3390/jpm12081306

**Published:** 2022-08-11

**Authors:** Iaroslav Skiba, Georgy Kopanitsa, Oleg Metsker, Stanislav Yanishevskiy, Alexey Polushin

**Affiliations:** 1Department of Chemotherapy and Stem Cell Transplantation for Cancer and Autoimmune Diseases, First Pavlov State Medical University of St. Peterburg, 197022 Saint Petersburg, Russia; 2Almazov National Medical Research Centre, 197341 Saint Petersburg, Russia; 3National Center for Cognitive Research, ITMO University, 49 Kronverskiy Prospect, 197101 Saint Petersburg, Russia

**Keywords:** oncohematology, risk factors, machine learning, epilepsy risk, epilepsy modeling

## Abstract

Machine learning methods to predict the risk of epilepsy, including vascular epilepsy, in oncohematological patients are currently considered promising. These methods are used in research to predict pharmacoresistant epilepsy and surgical treatment outcomes in order to determine the epileptogenic zone and functional neural systems in patients with epilepsy, as well as to develop new approaches to classification and perform other tasks. This paper presents the results of applying machine learning to analyzing data and developing diagnostic models of epilepsy in oncohematological and cardiovascular patients. This study contributes to solving the problem of often unjustified diagnosis of primary epilepsy in patients with oncohematological or cardiovascular pathology, prescribing antiseizure drugs to patients with single seizure syndromes without finding a disease associated with these cases. We analyzed the hospital database of the V.A. Almazov Scientific Research Center of the Ministry of Health of Russia. The study included 66,723 treatment episodes of patients with vascular diseases (I10–I15, I61–I69, I20–I25) and 16,383 episodes with malignant neoplasms of lymphoid, hematopoietic, and related tissues (C81–C96 according to ICD-10) for the period from 2010 to 2020. Data analysis and model calculations indicate that the best result was shown by gradient boosting with mean accuracy cross-validation score = 0.96. f1-score = 98, weighted avg precision = 93, recall = 96, f1-score = 94. The highest correlation coefficient for G40 and different clinical conditions was achieved with fibrillation, hypertension, stenosis or occlusion of the precerebral arteries (0.16), cerebral sinus thrombosis (0.089), arterial hypertension (0.17), age (0.03), non-traumatic intracranial hemorrhage (0.07), atrial fibrillation (0.05), delta absolute neutrophil count (0.05), platelet count at discharge (0.04), transfusion volume for stem cell transplantation (0.023). From the clinical point of view, the identified differences in the importance of predictors in a broader patient model are consistent with a practical algorithm for organic brain damage. Atrial fibrillation is one of the leading factors in the development of both ischemic and hemorrhagic strokes. At the same time, brain infarction can be accompanied both by the development of epileptic seizures in the acute period and by unprovoked epileptic seizures and development of epilepsy in the early recovery and in a longer period. In addition, a microembolism of the left heart chambers can lead to multiple microfocal lesions of the brain, which is one of the pathogenetic aspects of epilepsy in elderly patients. The presence of precordial fibrillation requires anticoagulant therapy, the use of which increases the risk of both spontaneous and traumatic intracranial hemorrhage.

## 1. Introduction

Malignant diseases of the hematopoietic system, despite their relatively low prevalence in the population, remain a socially significant group of diseases. Neurological complications in this cohort of patients occur in correlation with disease or with ongoing treatment. These complications may affect patient survival and may determine whether a therapy protocol can be fully implemented [1]. Acute symptomatic seizure (ASS) is one of the most significant neurological complications because of its high incidence and impact on survival [2]. A number of studies have evaluated the risk of ASS in this cohort of patients [3,4], while assessment of the risks of epilepsy is virtually unreported in the research to date [5,6].

Arterial hypertension is a cardiovascular complication in oncohematological patients that develops due to both disease-related and treatment-related factors [7,8]. Arterial hypertension has also been identified as one of the risk factors for the late-onset epilepsy in the general population [9].

Posterior reversible encephalopathy syndrome (PRES) is a brain disease associated with hypertension, which may determine the risk of epilepsy by indirect (in relation to arterial hypertension itself) mechanisms [10]. In the general population of patients with PRES syndrome, ACS occurs in 77% of cases [11]. In the cohort of oncohematological patients, the development of PRES syndrome may be accompanied by ASS in 97% of cases [12]. In the general population, arterial hypertension is the main etiological factor in the development of PRES syndrome (72%) [13]. It is a high-risk factor for the development of this complication in oncohematological patients as well (HR 14.466, 95% CI 7.107–29.443, *p* < 0.001) [12]. The risk of epilepsy in patients with PRES syndrome is considered low but may increase significantly in the presence of signs of cytotoxic edema and ASS in the debut of PRES syndrome [14].

The use of machine learning methods to predict the risk of complications in oncohematological patients has proven to be promising [15]. These methods are actively used in epilepsy, for example, to predict the pharmacoresistant epilepsy [16], to predict surgical treatment outcomes [17], to determine the epileptogenic zone [18] and to determine functional neural systems in patients with epilepsy [19], to develop new classification approaches [20,21], and to perform other tasks [22]. Machine learning models are actively used in decision support systems to treat patients with various forms of epilepsy [23,24]. At the same time, classical statistical methods of analysis are usually used to identify factors associated with the development of epilepsy within a typical case-control study design. However, factors related to the presence of epilepsy and prognostic tools that substantiate the optimal model for determining the risk of epilepsy in oncohematological patients are not fully understood now [25]. Currently, there exist no risk stratification models for epilepsy in oncohematological patients. The causes of symptomatic epilepsy are heterogeneous and require different approaches in the prevention of new foci of altered electrogenesis (e.g., brain infarcts in atrial fibrillation) [26].

The main goal of the study is to improve algorithms for diagnosing the cause of epilepsy in a group of patients without a previous history of epilepsy.

The groups of patients under consideration are patients with oncohematological diseases and cardiovascular pathology.

The main problem is the often-unjustified diagnosis of primary epilepsy in patients with oncohematological or cardiovascular pathology, prescribing antiseizure drugs to patients with single seizure syndromes without finding a disease associated with these episodes.

This paper presents the results of applying machine learning to analyzing data and developing diagnostic models of presence of epilepsy in oncohematological and cardiovascular patients.

We evaluate factors associated with the presence of epilepsy in oncohematological patients and the effect of arterial hypertension and the number of transplanted hematopoietic stem cells on the risk of epilepsy.

## 2. Materials and Methods

A single-center retrospective study was conducted. We analyzed the hospital database of the V.A. Almazov Scientific Research Center of the Ministry of Health of Russia. The study included 35,634 patients with 66,723 inpatient treatment cases (Dataset II) and 3723 patients with 16,383 inpatient treatment cases (Dataset I) of patients with malignant neoplasms of lymphoid, hematopoietic, and related tissues (C81–C96 according to ICD-10) for the period from the 27 January 2010 to the 5 January 2020. Laboratory parameters were chosen according to their clinical relevance and available data from real clinical practice. We considered their potential role in metabolism, systemic inflammation, and hemostasis and in the development of epileptic syndromes. Cerebrovascular factors were chosen according to the evidence on the increasing role of cardiovascular complications in predicting long-term outcomes in patients with oncohematological diseases.

### 2.1. Study Datasets

**Dataset I** was formed to develop a detailed descriptive and prognostic model of epilepsy for clinical, anamnestic, and laboratory patient factors in patients with oncohematology.


**Inclusion criteria**


Age: 1–90 years old;Diagnosis: verified malignant neoplasms of lymphoid, hematopoietic, and related tissues;Case type: inpatient treatment.


**Exclusion criteria**


Absence of oncohematological or cardiac disease. Outpatient treatment was an exclusion criterion;A history of Acute symptomatic seizures (ASS) without a verified diagnosis of epilepsy (G40.0–G40.8);Outpatient treatment.

A total of 356 factors were extracted for each patient, including genetic sex and constitutional factors, presence of comorbid pathology, factors for hematopoietic stem cell transplantation, and laboratory parameters.

The patients had the following clinical parameters:Comorbidities: 14% of I60–I69, fibrillation—6%, epilepsy (G40.0–G40.8)—1.5%, hypertension—20%;Genetic sex: females—49%, males—51%;Age: mean age—52.5 (min—1, max—90, 25%—40, 50%—57, 75%—66).

As an endpoint, we analyzed whether the patient developed epilepsy (presence of ICD-10 G40.0–G40.8 diagnoses).

**Dataset II** was formed to develop a descriptive and prognostic model of epilepsy in patients with cardiovascular disease to identify vascular factors in the development of epilepsy. Therefore, a group of patients with epilepsy both with and without oncohematological diagnosis was selected for this stage of the analysis to identify the contribution of the presence of oncohematological diagnosis to epilepsy. A second dataset was generated to analyze epilepsy in a wider patient group of 35,634 patients with 66,723 treatment episodes with 285 parameters among which: Age mean—55 (min—1, max 99, 25%—46, 50%—60, 75%—69), presence of comorbid diseases (hypertension, cerebral vascular disease, infarcts, atrial fibrillation and congenital heart disease (CHD), blood pressure, fibrillation (13%), G40—8%, males—44%, females—56%, BMI mean—1.87 (std—0.33, min—0.17, max—6.06, 25%—1.73, 50%—1.9, 75%—2.06).


**Inclusion criteria:**
Age: 1–99 years old;Diagnosis: hypertension, acute coronary syndrome (ACS), strokes, coronary artery disease (CAD),congenital heart disease (CHD), verified malignant neoplasms of lymphoid, hematopoietic, and related tissues;Case type: inpatient treatment.



**Exclusion criteria**
**:**
Absence of cardiovascular disease and oncological disease;Outpatient treatment was an exclusion criterion;A history of Acute symptomatic seizures (ASS) without a verified diagnosis of epilepsy (G40.0–G40.8).


The following data preparation procedure was performed for both datasets. We removed the patients with the insufficient amount of data (<80% of parameters). We also removed 1% of values having the highest z-score to filter out some obvious outliers.

After that, we applied two strategies of dealing with missing values to ensure that all patients have the same set of variables:

Replacement of missing data with the medians of the corresponding parameters;

Deletion of parameters that have too many missing values and removal of all patients that have any missing values in the remaining parameters.

### 2.2. Correlation Analysis

The Pearson coefficient was used to assess the correlation of the G40.0–G40.8 with the analyzed factors. The chi-squared criterion was applied to the binary values.

### 2.3. Machine Learning Methods

Gradient boosting and random forest models were applied.

Parameters of the model are XGBClassifier (base_score = 0.5, booster = None, colsample_bylevel = 1, colsample_bynode = 1, colsample_bytree = 0.6, gamma = 2, gpu_id = −1, importance_type = ‘gain’, interaction_constraints = None, learning_rate = 0.300000012, max_delta_step = 0, max_depth = 5, min_child_weight = 2, missing = nan, monotone_constraints = None, n_estimators = 100, n_jobs = 0, num_parallel_tree = 1, random_state = 0, reg_alpha = 0, reg_lambda = 1, scale_pos_weight = 1, subsample = 0.8, tree_method = None, validate_parameters = False, verbosity = None).

The parameters of the random forest were taken by default.

We also searched for optimal hyperparameters of the model using the greedy search method, Randomized Search CV, GridSearchCVwith params = { ‘min_child_weight’: [1, 2, 3, 4, 5, 6, 7, 8, 9, 10], ‘gamma’: [0.5, 1, 1.5, 2, 5]

‘subsample’: [0.1, 0.2, 0.3, 0.4, 0.5, 0.6, 0.8, 0.9, 1.0], ‘colsample_bytree’: [0.1, 0.2, 0.3, 0.4, 0.5, 0.6, 0.8, 0.9, 1.0], ‘max_depth’: [2, 3, 4, 5, 6, 7, 8, 9, 10, 15] }

We applied a stratified K-fold crossvalidation with 5 splits:

results = cross_val_score(rfc, X, y, cv = skf)

skf = StratifiedKFold(n_splits = 5, shuffle = True, random_state = 42)

The experiments were performed using the following hardware:

Intel Core i3-8109U CPU (3.00 GHz);

8 GM of Ram;

64-bit Windows 10 operating system.

The average times of the experiments were:

Dataset I: 01 min 19 s;

Dataset II: 52 min 18 s.

### 2.4. Importance of Predictors

Using the Shapley index, predictor significance factors were calculated in a model on an epilepsy class in patients with oncohematology and in a sample of patients with cerebrovascular disease.

### 2.5. Cerebrovascular Disease

After the first stage of data analysis, we detailed the parameters of cerebrovascular pathology I60–I69 as a significant factor associated with the presence of epilepsy. The detailing was carried out according to ICD-10 classification subheadings and included a search for specific nosological forms, including that within I67 and I65nosologies, which showed the greatest significance in the model for the “epilepsy” class.

The ranking of predictors for the presence of an epilepsy diagnosis on Dataset II was performed using the built-in method of predictor significance according to the Gini criterion in sklearn (also known as skikit-learn) with the setting of “balanced_subsample” weights autobalancing. The weight of each subsample varied according to the class distribution in that subsample. The built-in method provided an estimate of the significance of each individual feature in the model, in contrast to the Shapley index.

## 3. Results

Demographic details of the patients are presented in the Table 1.

### 3.1. Dataset I

As a result of data analysis and model calculations, the best result was shown by gradient boosting (Table 2) with mean accuracy cross-validation score = 0.96, f1-score = 0.98, weighted avg precision = 0.93, recall = 0.96.

The highest correlation coefficient for the presence of epilepsy and recurrent seizures (G40) was achieved with stenosis or occlusion of the precerebral arteries (0.16), cerebral sinus thrombosis (0.089), arterial hypertension (0.17), age (0.03), non-traumatic intracranial hemorrhage (0.07), atrial fibrillation (0.05), delta absolute neutrophil count (0.05), platelet count at discharge (0.04), transfusion volume for stem cell transplantation (0.023). The discriminative ability of the model calculated as AUC of ROC is 0.94.

The results of the Shapley index analysis of factors associated with the development of epilepsy in oncohematological patients are presented in the Figure 1 that contains the following predictors:

Hypertension—arterial hypertension;

I65—Occlusion and stenosis of precerebral arteries, not resulting in cerebral infarction;

MON—monocytes, maximum absolute number;

MCH—average concentration of hemoglobin in the erythrocyte;

NEUT—absolute neutrophil count;

BMI—body mass index;

MCV—mean erythrocyte volume;

PLT—platelets;

MCH min—minimum mean content of hemoglobin in an erythrocyte;

BMI max—maximum body mass index;

I67.6—Nonpyogenic thrombosis of intracranial venous system;

HCT—hematocrit;

NEUT otn—neutrophils, fraction in %;

PLT max—maximum platelet count;

Na—natrium;

I69—Sequelae of cerebrovascular disease;

I67.0—Other cerebrovascular diseases;

PLT average y—Platelets, mean per year;

Na average—mean sodium;

Bood type—blood group;

Quantity of transplanted cells—Quantity of transplanted cells;

I66—Occlusion and stenosis of cerebral arteries, not resulting in cerebral infarction;

I64—Stroke, not specified as hemorrhage or infarction;

Lym_abs_max—absolute lymphocyte count;

Female-woman—Female gender;

PLT_first_x—platelets, first measurement;

I67.1–Cerebral aneurysm, nonruptured;

MCH_first_y—Mean concentration of hemoglobin in an erythrocyte in the first blood test;

MCH_average_y—Mean concentration of hemoglobin in an erythrocyte

I63–Cerebral infarction;

MCH_max_y—maximum mean concentration of hemoglobin in the erythrocyte;

RDW—erythrocyte size distribution;

I67.5—Moyamoya disease;

I68—Cerebrovascular disorders in diseases classified elsewhere.

The effect of cerebral venous sinus thrombosis and arterial hypertension on the risk of epilepsy is shown in Figure 2.

The effect of the number of transplanted hematopoietic stem cells on the risk of epilepsy is shown in Figure 3.

### 3.2. Dataset II

For the next stage of the analysis, a group of patients with epilepsy with and without the oncohematological diagnosis was selected to identify the contribution of the presence of oncohematological diagnosis on epilepsy (see Table 3 for the evaluation).

Figure 4 shows that children (1–17 years old) are characterized by a slight increase in the risk of epilepsy in the presence of oncohematological disease, but the absence of oncological disease significantly increases the risk of epilepsy.

Figure 5 demonstrates the plot of dependence of oncohematological diagnosis on the presence of epilepsy and age.

In the age group of 18–40 years old, there was a gradual decrease in the influence of the “oncohematological disease” factor on the risk of epilepsy (Figure 6). In the age range of 40–60 years old, the presence/absence of oncohematological disease had almost no effect on the risk of epilepsy. At the same time, after the age of 60, the presence of such disease increased the risk of epilepsy in patients.

When ranking the features associated with the presence of epilepsy, atrial fibrillation was found to have the highest weight, and oncohematological disease was the third most important feature (Figure 6).

## 4. Discussion

### 4.1. General

A number of factors (clinical, gender, and laboratory) have been associated with the development of epilepsy in oncohematological patients. These factors can be grouped as follows:
vital signs (age, body mass index, patient weight);cardiovascular pathology, cerebrovascular pathology (arterial hypertension, stenosis or occlusion, occlusion and stenosis of precerebral arteries, cerebral sinus thrombosis, cerebral artery dissection without rupture, cerebral aneurysmatic disease, cerebral infarction);laboratory parameters (maximum absolute monocyte count, average hemoglobin content of red blood cells, neutrophil count, platelet count at hospital discharge, minimum hematocrit value, minimum and average blood sodium levels);hematopoietic stem cell transplantation parameters (donor blood group, number of transplanted cells).

A number of factors had no significant effect on the risk of epilepsy. Demographic characteristics such as sex, age, and body weight were generally not significant. In spite of the known features of the age distribution of epilepsy, this relationship was not detected in the group of oncohematological patients. This fact may be related to the peculiarities of etiological factors in this cohort of patients, but further studies are needed to clarify the influence of various factors in different age groups.

The inclusion criteria in the first stage were limiting patients with oncohematological pathology and checking that possible structural or functional brain changes in oncohematological pathology would be associated with the development of epilepsy (more for secondary epilepsy).

In fact, there was a study of oncology-epilepsy and a high incidence of epilepsy was confirmed. The inclusion of patients with cardiovascular diseases in the second stage expanded the patient base due to a higher incidence of cardiovascular diseases, and the rank of nosological form (specific disease) in the development of comorbid epilepsy was determined.

### 4.2. Study Population

In the first dataset, the proportions of males and females were comparable, and the incidence of epilepsy was 1.5%. This prevalence of epilepsy in the first dataset was significantly higher than in the general population (1%) [27]. The incidence of atrial fibrillation (6%) at a mean age of 52.5 years (lower in comparison to other studies in the general population) was also comparable to that in the general population [28], including in the Russian Federation (5.8% for men and 7.4% for women) [29]. In the second dataset, the prevalence of atrial fibrillation was more than twice as high as in the first dataset. This may be associated with high comorbidity of this variant of rhythm disturbance and cardiovascular diseases [30]. Despite the lower proportion of males in dataset II (male gender is generally associated with a higher incidence of epileptic syndromes) [31]. The prevalence of epilepsy was significantly higher (8%) than in the first dataset.

### 4.3. Risk Factors

Among the laboratory parameters, both granulocytic and erythrocytic hematopoiesis parameters and platelet levels were found to be significant factors influencing the presence of epilepsy in oncohematological patients. Changes in these parameters are influenced by both the blood disease itself and its complications, as well as the administered therapy. A higher neutrophil count may indirectly indicate infectious complications or a more severe/prolonged course. In this regard, we can assume that factors such as infectious complications and hypofunction of the transplant, which are accompanied by significant laboratory changes, may explain the presence of these factors among the parameters influencing the epilepsy in this group of patients. Thrombocytopenia may be a risk factor for primarily intracranial hemorrhages in oncohematological patients [32,33,34]. These complications can lead to the formation of an epileptogenic substrate in the brain. In addition, the lower platelet count could be due to the antiseizure therapy being taken. Although epidemiological data on the development of epilepsy in patients with cerebral sinus thrombosis are not available, our results concerning the role of this factor in the development of epilepsy are more significant. Unfortunately, extrapolation to the general population is not possible given the sample of patients in our study (oncohematological patients). Thrombocytopenia was not previously considered a risk factor for epilepsy in the general population, as well as in the population of patients with other neurological [35,36,37] or general pathology [38].

The relationship between the development of cerebral venous sinus thrombosis (CVST) and oncology in general, and oncohematology in particular, has been investigated in a number of scientific papers [39]. Patients with oncohematologic diseases have a higher risk of CVST (aOR, 25.14; 95% CI, 11.64–54.30) than patients with solid cancer (aOR, 3.07; 95% CI, 2.03–4.65) [40]. Risks are even more severe in the first year after the cancer has been diagnosed (oncohematological OR, 85.57; 95% CI, 19.70–371.69; solid cancer aOR, 10.50; 95% CI, 5.40–20.42) [40]. CVST can account for up to 31.5% of all venous thromboembolic complications, for example, in a cohort of adult patients with acute myeloblastic leukemia [41].

Dimethyl sulfoxide as a factor is considered in the studies and can provoke the development of ASS. It can cause cardiovascular complications including ischemic stroke in the early post-transplant period [42,43,44]. In this regard, it is important to emphasize the contribution of the number of cells injected during transplantation in the risk of epilepsy that we identified in patients with arterial hypertension. This emphasizes the possible cardiovascular mechanisms of this effect.

The significance of the factor of presence of I67.6 (Nonpyogenic thrombosis of intracranial venous system) including in young patients could be due to indirect mechanisms of stroke development. Epidemiological data on the development of epilepsy in patients with cerebral sinus thrombosis are not available. This emphasizes the significance of our findings regarding the role of this factor in the development of epilepsy.

### 4.4. Arterial Hypertension

Arterial hypertension contributed the most to the presence of epilepsy in oncohematological patients (Figure 1). This, along with the presence of less significant vascular factors (extracranial arteries stenosis/occlusion, cerebral sinus thrombosis, cerebral artery dissection without rupture, cerebral aneurysmatic disease, cerebral infarction), may indicate a significant role of cerebrovascular pathology in the presence of epilepsy in this group of patients [45]. Younger age may be related to a known peak of higher prevalence of epilepsy specifically in young people in the general population [46]. This fact may be associated with a higher incidence of complications associated with arterial hypertension at a young age, for example, PRES. The presence of arterial hypertension in younger patients (18–23 years old) significantly increased the likelihood of epilepsy, while in older patients (including the elderly, over 65 years of age), its presence had the opposite effect: it reduced the likelihood of epilepsy.

Interestingly, the combined presence or absence of cerebral venous sinus thrombosis and arterial hypertension altered the likelihood of epilepsy in patients.

### 4.5. Cerebral Sinus Thrombosis

As shown in the Figure 7, in the absence of Cerebral venous sinus thrombosis (CVST), the presence/absence of arterial hypertension had no significant effect on the risk of epilepsy. While during the development of CVST, arterial hypertension sharply increased its significance as a factor in the presence of epilepsy in the patient.

CVST itself is a well-known risk factor for the development of acute symptomatic epileptic seizures and epilepsy [47,48,49]. Meanwhile, the relationship between CVST and arterial hypertension in terms of epilepsy risk has been described for the first time and may reflect the presence of combined mechanisms of these conditions in oncohematological patients.

A number of hematologic factors can determine the risk of CVST. The development of this complication may be associated with a higher platelet count (*p* < 0.001) and a higher platelet/neutrophil index (*p* < 0.001) [50]. The incidence of ASS in the acute phase of CVST is up to 34% in the general adult population. The incidence of ASS in CVST in children is 37.5% to 57%. This is often the main manifestation of the development of thrombosis [51,52]. The results of a study of the risk of epilepsy in young patients and children show that systemic inflammation, a reflection of which may be the fact of increased platelets, may play a significant role. This role can be both direct and mediated through the development of complications involving the CNS) in the development of epilepsy [53]. Given the heterogeneity of causes leading to the development of elevated neutrophil levels, unambiguous interpretation is difficult and the role of systemic inflammation in the development of epilepsy requires further study.

### 4.6. Transplanted Hematopoietic Stem Cells

Among the factors associated with hematopoietic stem cell transplantation, a higher volume of transplanted cells was associated with the presence of epilepsy (Figure 4). This factor increased its weight in patients with arterial hypertension. This may indicate a possible influence of the volume of a donor cell infusion, and the volume of injected cryopreservative on the development acute arterial hypertension and other complications, including ASS [54].

### 4.7. Age Factor

The factor of age for predicting the presence of epilepsy in oncohematological patients had different significance depending on the age group (Figure 5). In patients under 18 years old, the presence of a malignant neoplasm of the blood system reduced the likelihood of epilepsy. This may be related to the debut of hereditary genetically determined forms of epilepsy syndromes with no etiological and pathogenetic connection to oncohematological diseases [55]. At the age of 18–20 years, there was an increase in the prognostic significance of the presence of C81–C96 on the risk of epilepsy. This may be related to the transition of patients to adult inpatient care, changes in neoplasm treatment protocols, and possible disruption of continuity between specialists [56]. This leads to a possible increase in the risk of complications. In the age group of 60 years and older, the presence of C81–C96 diseases increased the likelihood of a patient having epilepsy. This is associated with an increased number of complications leading to damage of the brain substance due to the presence of other comorbid pathology (primarily, pathology of the cardiovascular system) and its decompensation against the background of oncological disease therapy [55].

At the same time, in the same sample of patients, when analyzing the importance of predictors for the oncohematological diagnosis class, we can see that the absence of epilepsy is in no way related and does not contribute to the model for the oncohematological diagnosis class, as opposed to the presence of epilepsy (Figure 6).

### 4.8. Dataset I vs. Dataset II Patients

When performing a features-engineering of a model of epilepsy in oncohematological patients, a model development cycle should include a step to compare the importance of the features with the model that considers a wider group of patients. If differences in the importance of predictors are found, validation and interpretation of the results, and adjustment of the initial narrow model with the identified limitations are necessary.

From the clinical point of view, the identified differences in the importance of predictors (Figure 7) in a broader patient model are consistent with a practical algorithm for organic brain damage. Atrial fibrillation is one of the leading factors in the development of both ischemic and hemorrhagic strokes. At the same time, brain infarction can be accompanied both by the development of epileptic seizures in the acute period and by unprovoked epileptic seizures and development of epilepsy in the early recovery and in a longer period [57]. In addition, microembolism of the left heart chambers can lead to multiple microfocal lesions of the brain, which is considered to be one of the pathogenetic aspects of epilepsy in elderly patients. The presence of precordial fibrillation requires anticoagulant therapy, the use of which increases the risk of both spontaneous and traumatic intracranial hemorrhage. Which, in the case of involvement of the brain substance, forms an epileptogenic substrate in the form of hemosiderin zones [58].

### 4.9. Clinical Implications

Alongside identifying of the risk factors and the ability of the model to rate the risks of individual patients, our results have other practical clinical meaning. It is important that in patients with a cardiovascular history, the development of seizures is not always associated with primary epilepsy. It is structural changes in the brain of acute or chronic genesis can be inducers of seizures and secondary epilepsy. Moreover, given that the most frequent diagnosis of “epilepsy” was eventually found in patients with atrial fibrillation, in the absence of electroencephalogram changes and the presence of sinus rhythm on electrocardiogram, and the absence of gross structural changes in the brain, long-term cardiac rhythm monitoring should be conducted to look for a paroxysmal form of atrial fibrillation.

From a clinical point perspective, our results may be useful for classifying and ranking the causes of epileptic seizures, especially in the group of patients with no history of epilepsy as the primary pathology. The developed prognostic model made it possible to identify factors associated with epilepsy in patients with oncohematological diseases. Further study of the causal relationship between these factors and the development of epilepsy will allow for the creation of an algorithm for its timely prevention.

### 4.10. Study Limitation

The limitations of this study are related to the neurotoxicity of a number of chemotherapy drugs used to treat oncohematological patients, which increase the risk of ASS (acute symptomatic seizure) and other neurological complications. This can lead to structural damage to the brain substance (e.g., ischemic stroke and intracranial hemorrhage).

The inclusion and exclusion criteria we selected and the parameters chosen for analysis did not take into account the stage of the disease and the protocol of the patient’s therapy, nor did we consider the reason for the patient’s admission to the hospital and its urgency.

In our study, we considered the presence of epilepsy, coded as a competing, comorbid disease or complication, in patients with oncohematological (dataset 1) and cardiovascular diseases (dataset 2) during the contraction time (period of admission); however, the fact of epilepsy in dynamics and the risk factors of such an event, were not assessed. We focused on finding factors associated with the presence of epilepsy rather than its development (predictors of its development) in the future.

## 5. Conclusions

Patients with oncohematological pathology have a number of clinical and laboratory factors that correlate with the presence of epilepsy. In addition, age and a number of factors associated with hematopoietic cell transplantation also correlate with the presence of epilepsy in patients with malignant neoplasms of lymphoid, hematopoietic and related tissues. This study shows how age characteristics influence the significance of other predictors. Patient age and shelf life of the transplanted cells change the significance. The obtained results of the thrombosis factor in the development of epilepsy have a meaningful effect because this issue has been poorly described in previous studies. Further research should be aimed at creating a prognostic model for the development of ASS in this group of patients. We proposed to include ASS as a risk factor to the prognostic model of epilepsy development in oncohematological patients. In addition, the identified association between platelet levels and epilepsy requires a study of the effect of the use of antiepileptic drugs in patients with malignant blood neoplasms.

## Figures and Tables

**Figure 1 jpm-12-01306-f001:**
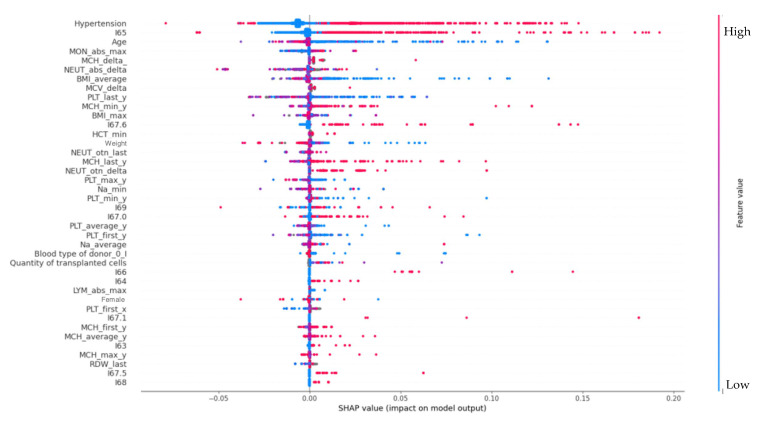
Factors associated with the development of epilepsy in oncohematological patients.

**Figure 2 jpm-12-01306-f002:**
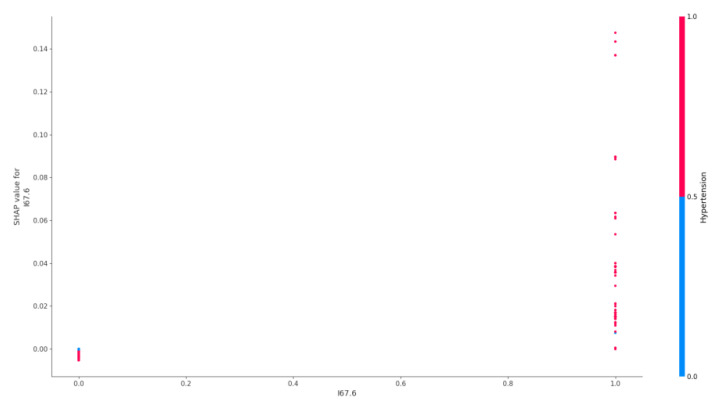
Effect of cerebral venous sinus thrombosis and arterial hypertension on the risk of epilepsy.

**Figure 3 jpm-12-01306-f003:**
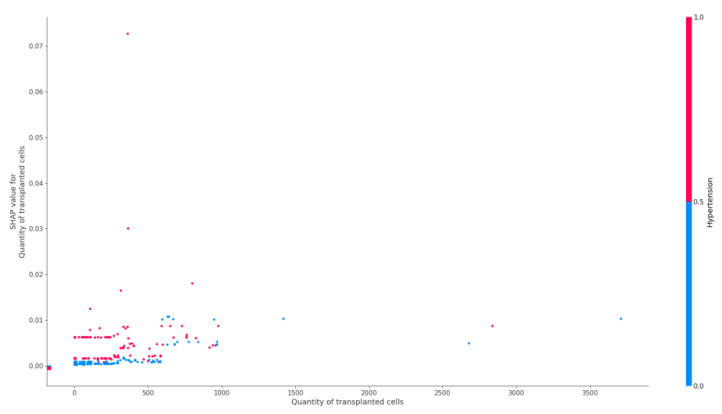
Dependency of the number of transplanted hematopoietic stem cells on hypertension in the model of an epilepsy class.

**Figure 4 jpm-12-01306-f004:**
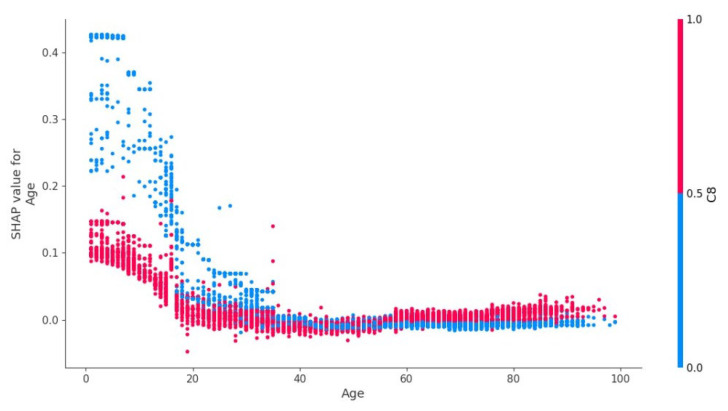
Dependency of epilepsy on age and presence of oncohematological diagnosis.

**Figure 5 jpm-12-01306-f005:**
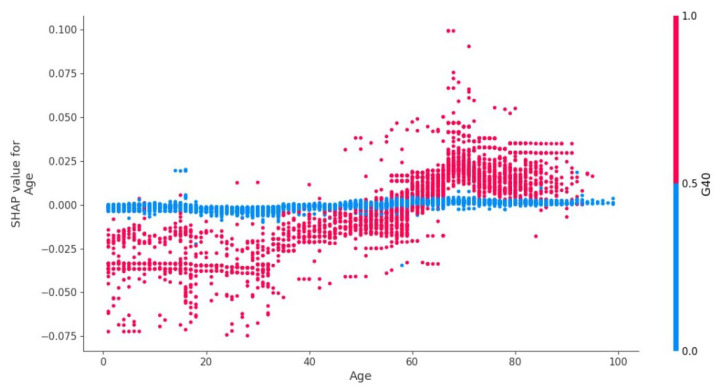
Dependency of oncohematological diagnosis on the presence of epilepsy and age.

**Figure 6 jpm-12-01306-f006:**
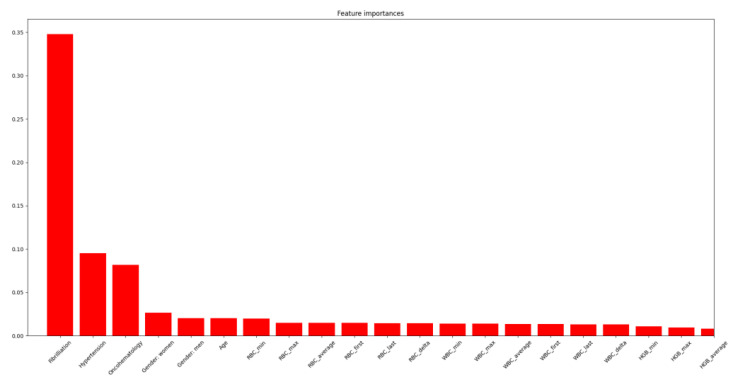
Predictor ranking for the onset of epilepsy in Dataset II.

**Figure 7 jpm-12-01306-f007:**
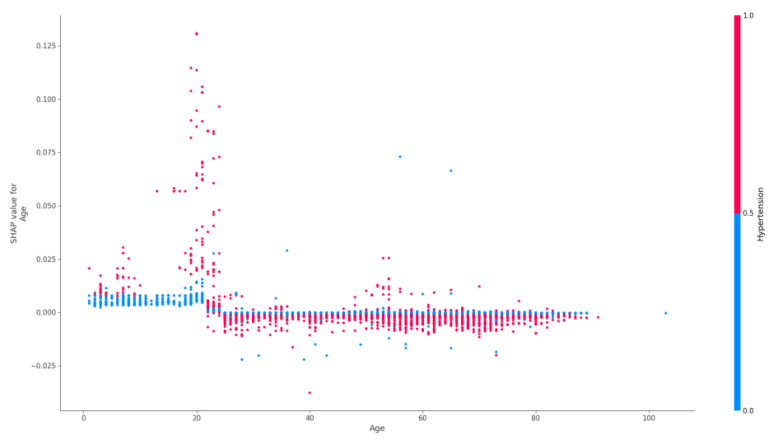
Effect of arterial hypertension on the risk of epilepsy depending on the age of patients.

**Table 1 jpm-12-01306-t001:** Demographic details of the study population.

Dataset	Males	Females	Mean Age	Age 25%	Age 50%	Age 75%	Comorbidities
Dataset I	51%	49%	52.5	40	57	66	14% of I60–I69, Fibrillation—6%, epilepsy (G40.0–G40.8)—1.5%, hypertension—20%
Dataset II	44%	56%	55	46	60	69	presence of comorbid diseases (hypertension, cerebral vascular disease, infarcts, atrial fibrillation and congenital heart disease (CHD), blood pressure, fibrillation (13%), G40—8%

**Table 2 jpm-12-01306-t002:** Model evaluation for the Dataset I.

Method	Accuracy	Precision	Recall	F1-Score	AUC of ROC
Gradient Boosting	0.96	0.93	0.96	0.98	0.94
Random forest	0.92	0.89	0.93	0.94	0.91

**Table 3 jpm-12-01306-t003:** Model evaluation for the Dataset II.

Method	Cross-Validation Score	Precision	Recall	F1-Score	AUC of ROC
Gradient Boosting	0.93	0.91	0.94	0.94	0.94
Random forest	0.89	0.82	0.91	0.90	0.90

## Data Availability

The datasets used and/or analysed during the current study are available from the corresponding author on reasonable request.

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
