# Peer review of "Application of Machine Learning Methods for Epilepsy Risk Ranking in Patients with Hematopoietic Malignancies Using"

_jpm, 2022, doi:10.3390/jpm12081306_

Round 1
Reviewer 1 Report
· I do not see authors using any data cleaning approaches, if used please explicitly mention those.
· The applications of the results (including experiments) generated by the proposed approach are not clearly explained. Who will benefit from the results?
· How to evaluate the accuracy of the proposed approach? Did you do any expert evaluation?
Author Response
Q1 I do not see authors using any data cleaning approaches, if used please explicitly mention those.
We added data cleaning methods to the paper:
The following data preparation procedure was performed for both datasets. We removed the patients with the insufficient amount of data (< 80% of parameters). We also removed 1% of values having the highest z-score to filter out some obvious outliers.
After that, we applied two strategies of dealing with missing values to ensure that all patients have the same set of variables:
Replacement of missing data with the medians of the corresponding parameters;
Deletion of parameters that have too many missing values and removal of all patients that have any missing values in the remaining parameters.
Q2 The applications of the results (including experiments) generated by the proposed approach are not clearly explained. Who will benefit from the results?
The following ws added to the discussion, section 4.9
From a clinical point perspective, our results may be useful for classifying and ranking the causes of epileptic seizures, especially in the group of patients with no history of epilepsy as the primary pathology.
Q3 How to evaluate the accuracy of the proposed approach? Did you do any expert evaluation?
We didn't do the an exper evaluation. The results were clinically interpreted by the doctors. The technical validity of the approach was evaluated using accuracy, recall, f-measure and AUC of ROC meaurements. Please see table 2.
Reviewer 2 Report
This paper applies gradient boosting/random forest to analyzing two hospital datasets and diagnostic features that contribute to the presence of epilepsy in oncohematological and cardiovascular patients.
Some comments:
The authors state ‘Development of epilepsy risk ranking models in patients with hematopoietic malignancies using machine learning’, ‘...developing diagnostic models of presence of epilepsy in oncohematological and cardiovascular patients’, .. . The word ‘development/developing’ is not appropriate because this is an applied paper and the gradient boosting, random forest, Pearson coefficient, Shapley index, Gini criteria used in this paper are all well-developed methods. Citation/reference need to be provided and a one or two sentences description of those methods would be appreciated by readers without such background.
F1-score, recall, precision, etc should be a ratio between 0-1.
The authors state gradient boosting outperformed random forest on dataset I. The random forest results should be listed for comparison.
Did the gradient boosting algorithm also outperform random forest on dataset II? Was the discussion of dataset II based on gradient boosting results?
In the conclusion section, after summarizing the findings of the paper, what further study can be conducted based on the paper’s results? What are some new perspectives that this paper inspired? Including more contributions/potential future studies would increase the impact of the paper.
Some figure x, y labels are too small and hard to read
Author Response
Q1 : The authors state ‘Development of epilepsy risk ranking models in patients with hematopoietic malignancies using machine learning’, ‘...developing diagnostic models of presence of epilepsy in oncohematological and cardiovascular patients’, .. . The word ‘development/developing’ is not appropriate because this is an applied paper and the gradient boosting, random forest, Pearson coefficient, Shapley index, Gini criteria used in this paper are all well-developed methods. Citation/reference need to be provided and a one or two sentences description of those methods would be appreciated by readers without such background.
We changed the title to:
Application of machine learning methods for epilepsy risk ranking in patients with hematopoietic malignancies using
Q2 F1-score, recall, precision, etc should be a ratio between 0-1.
This was corrected in the text, please see table 2
Q3 The authors state gradient boosting outperformed random forest on dataset I. The random forest results should be listed for comparison.
Please see table 2 for the comparison
Q4 Did the gradient boosting algorithm also outperform random forest on dataset II? Was the discussion of dataset II based on gradient boosting results?
Please see table 3 for the comparison
Q5 In the conclusion section, after summarizing the findings of the paper, what further study can be conducted based on the paper’s results? What are some new perspectives that this paper inspired? Including more contributions/potential future studies would increase the impact of the paper.
Further research should be aimed at creating a prognostic model for the development of ASS in this group of patients. We proposed to include ASS as a risk factor to the prognostic model of epilepsy development in oncohematological patients. In addition, the identified association between platelet levels and epilepsy requires a study of the effect of the use of antiepileptic drugs in patients with malignant blood neoplasms.
Q6 Some figure x, y labels are too small and hard to read
The labels were corrected
Reviewer 3 Report
The reviewer doubts that these results can be generalized. The comparison methods need to be chosen appropriately and recent. The comparative study from the recently proposed approach is missing. However, at this moment, the novelty of the proposed approach is limited, and several technical details are missing, some specific points need to be noted or improved significantly. Its current status cannot reach the level of this journal for publication.
1- Please explain the validation method in more detail.
Please report the accuracy of the proposed method by Leaving one subject out of validation (subject-wise CV). Most expert diagnosis systems were validated in recent years based on the leave-one subject out. The record-wise CV randomly splits data into training and test, which randomly splits data into training and test sets regardless of which subjects they belong to. Therefore, records from the same subject are present in both training and test sets. In this way, the machine learning algorithm can find an association between unique features of a subject and their state, which automatically improves its prediction accuracy on their test data. Consequently, the record-wise CV method can significantly overestimate the predicted accuracy of the algorithm. This biased estimation may occur in record-wise CV due to shared identities, while subject-wise CV calculates an out-of-sample prediction error.
2- The authors should describe their proposed system in more detail. The reviewer suggested adding a flowchart to represent the system better. The authors’ framework is simple and common sense, and the detailed methods used in each module are also off-the-peg. I cannot see what the true contributions are.
3- Please improve the quality of the figures; they should be readable.
4- The authors need to explain how the method helps the current field and what challenges it solves.
5- The existing derivation process is extensively shown in the paper, and more explanation is needed on how to design the method to tackle the challenges.
6- A simpler and more efficient visualization can be provided.
7- It can be properly explained in the text when it is displayed visually.
8- In the Introduction, the manuscript does not clearly indicate the advantages and innovations of the proposed method. The differences from the existing work need to be explained in detail.
9- What about the computational complexity (e.g., running time analysis)?
10- The caption of figures and tables should be more informative.
11- For more impact on the manuscript and validation, sharing the proposed neural network codes on GitHub is better.
12- What makes the proposed method suitable for this unique task? What new development to the proposed method have the authors added (compared to the existing approaches)? These points should be clarified. The discussions should highlight why the proposed method is providing good results. The comparative study from the recently proposed system is missing.
13- Please explain in all detail how the authors selected the parameters?
Author Response
Q1 Please explain the validation method in more detail.
Please report the accuracy of the proposed method by Leaving one subject out of validation (subject-wise CV). Most expert diagnosis systems were validated in recent years based on the leave-one subject out. The record-wise CV randomly splits data into training and test, which randomly splits data into training and test sets regardless of which subjects they belong to. Therefore, records from the same subject are present in both training and test sets. In this way, the machine learning algorithm can find an association between unique features of a subject and their state, which automatically improves its prediction accuracy on their test data. Consequently, the record-wise CV method can significantly overestimate the predicted accuracy of the algorithm. This biased estimation may occur in record-wise CV due to shared identities, while subject-wise CV calculates an out-of-sample prediction error.
We applied a stratified K-fold crossvalidation with 5 splits:
results = cross_val_score(rfc, X, y, cv=skf)
skf = StratifiedKFold(n_splits=5, shuffle=True, random_state=42)
Q2 The authors should describe their proposed system in more detail. The reviewer suggested adding a flowchart to represent the system better. The authors’ framework is simple and common sense, and the detailed methods used in each module are also off-the-peg. I cannot see what the true contributions are.
We did not have a goal to develop a system. We developed a series of machine learning models. They can be integrated in the clinical decision support systems in the future. Our contribution is presented in teh discussion section
Q3 Please improve the quality of the figures; they should be readable.
The quality has been improved
Q4 [Doctors] The authors need to explain how the method helps the current field and what challenges it solves.
We added the following to the discussion section:
The developed prognostic model made it possible to identify factors associated with epilepsy in patients with oncohematological diseases. Further study of the causal relationship between these factors and the development of epilepsy will allow to create an algorithm for its timely prevention.
Q5 The existing derivation process is extensively shown in the paper, and more explanation is needed on how to design the method to tackle the challenges.
We applied cross-validation. Startidief k-fold scilearn library.
results = cross_val_score(rfc, X, y, cv=skf)
skf = StratifiedKFold(n_splits=5, shuffle=True, random_state=42)
Q6 A simpler and more efficient visualization can be provided.
We provided all the required visualization of the results. Figures 1-7 fully represent the results of the application of machine learning methods.
Q7 In the Introduction, the manuscript does not clearly indicate the advantages and innovations of the proposed method. The differences from the existing work need to be explained in detail.
However, factors related to the presence of epilepsy and prognostic tools that substantiate the optimal model for determining the risk of epilepsy in oncohematological patients are not fully understood now [27]. Currently, there exist no risk stratification model for epilepsy in oncohematological patients. The causes of symptomatic epilepsy are heterogeneous and require different approaches in the prevention of new foci of altered electrogenesis (e.g. - brain infarcts in atrial fibrillation) [28].
Q8 What about the computational complexity (e.g., running time analysis)?
The experiments were performed using the following hardware:
Intel Core i3-8109U CPU (3.00 GHz)
8 GM of Ram
64 bit Windows 10 operating system
The average times of the experiments were:
Dataset I: 01 minute 19 seconds
Dataset II: 52 minutes 18 seconds
Q10 The caption of figures and tables should be more informative.
The captions were improved to provide more information
Q11 For more impact on the manuscript and validation, sharing the proposed neural network codes on GitHub is better.
We added a github link:
https://github.com/OlegMetsker/paper/blob/main/epilepsy
Q12 What makes the proposed method suitable for this unique task? What new development to the proposed method have the authors added (compared to the existing approaches)? These points should be clarified. The discussions should highlight why the proposed method is providing good results. The comparative study from the recently proposed system is missing.
The detailed description of what makes the proposed methods suitable for this unique task is given in the discussion section with clinical interpretation of the results and implication for future clinical practice. As the study is based on the unique dataset with patients with hematopoietic malignancies, it would be hard and not efficient to compare the technical accuracy of the proposed methods with other studies.
Q13 Please explain in all detail how the authors selected the parameters?
Laboratory parameters were chosen according to their clinical relevance and available data from real clinical practice. We considered their potential role in metabolism, systemic inflammation, and hemostasis and in the development of epileptic syndromes. Cerebrovascular factors were chosen according to the evidence on the increasing role of cardiovascular complications in predicting long-term outcomes in patients with oncohematological diseases.
Round 2
Reviewer 3 Report
With the corrections made, the paper seems acceptable to me for publication in the journal.